# BadFusion: 2D-Oriented Backdoor Attacks against 3D Object Detection

**Saket Sanjeev Chaturvedi[1], Lan Zhang[1], Wenbin Zhang[2], Pan He[3], Xiaoyong Yuan[1]**
[1]Michigan Technological University
[2]Florida International University    [3]Auburn University

## Abstract

3D object detection plays an important role in autonomous driving; however, its vulnerability to backdoor attacks has become evident. By injecting "triggers" to poison the training dataset, backdoor attacks manipulate the detector's prediction for inputs containing these triggers. Existing backdoor attacks against 3D object detection primarily poison 3D LiDAR signals, where large-sized 3D triggers are injected to ensure their visibility within the sparse 3D space, rendering them easy to detect and impractical in real-world scenarios. In this paper, we delve into the robustness of 3D object detection, exploring a new backdoor attack surface through 2D cameras. Given the prevalent adoption of camera and LiDAR signal fusion for high-fidelity 3D perception, we investigate the latent potential of camera signals to disrupt the process. Although the dense nature of camera signals enables the use of nearly imperceptible small-sized triggers to mislead 2D object detection, realizing 2D-oriented backdoor attacks against 3D object detection is non-trivial. The primary challenge emerges from the fusion process that transforms camera signals into a 3D space, thereby compromising the association with the 2D trigger to the target output. To tackle this issue, we propose an innovative 2D-oriented backdoor attack against LiDAR-camera fusion methods for 3D object detection, named BadFusion, aiming to uphold trigger effectiveness throughout the entire fusion process. Extensive experiments validate the effectiveness of BadFusion, achieving a significantly higher attack success rate compared to existing 2D-oriented attacks.

## 1 Introduction

3D object detection has become a core component for many state-of-the-art autonomous driving systems Qian et al. [2022]. By accurately recognizing and localizing objects like vehicles, pedestrians, and cyclists, 3D object detection enhances the ability of driving systems to perceive and understand surroundings, enabling them to make responsible decisions. Despite the significant progress achieved by deep neural networks in 3D object detection, it has been demonstrated that neural network-based object detectors are susceptible to backdoor attacks Xiang et al. [2021], Li et al. [2021a], Zhang et al. [2022]. Backdoor attackers contaminate the detector's training dataset by injecting "triggers," which consequently mislead predictions during inference. The prevalence of backdoor attacks poses significant safety hazards, particularly in safety-critical driving scenarios.

Existing backdoor attacks against 3D object detection mainly inject triggers to LiDAR signals because the spatial information provided by LiDAR offers critical 3D detection evidence. However, due to the sparsity of LiDAR signals in most commercialized LiDAR sensors, backdoor attacks require adding large-size triggers to the target vehicle to ensure that the trigger information can be effectively captured. For example, Zhang et al. Zhang et al. [2022] used a cargo carrier bag with a size of 1.1m $\times$0.8m $\times$0.5m or an exercise ball with a radius of 0.4m as a trigger, which is mounted on the roof of the target vehicle for backdoor attacks. Such large 3D triggers can significantly change the vehicle's shape and appearance and thus be easily detected, making 3D backdoor attacks impractical to implement in real-world scenarios. Therefore, to thoroughly investigate the robustness of 3D object detection, in this paper, we explore a new potential attack surface through 2D camera signals.

Published at NeurIPS 2023 Workshop on Backdoors in Deep Learning: The Good, the Bad, and the Ugly.

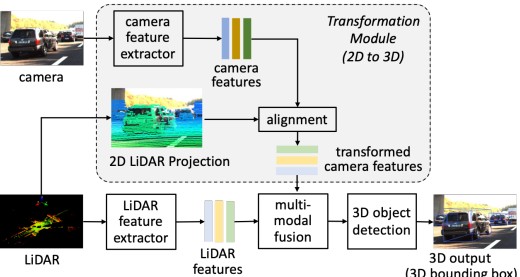

Figure 1: The pipeline of 2D (camera) and 3D (LiDAR) data fusion for 3D object detection in autonomous driving.

In addition to LiDAR signals, camera signals have been another prominent source of input for 3D object detection. Compared to 3D spatial yet low-resolution signals from LiDAR, cameras capture high-resolution color features, yielding robust fusion outcomes that significantly enhance the quality of 3D perception Wang et al. [2021], Yin et al. [2021], Li et al. [2022]. However, the popularity of these multi-modal systems leads to a new backdoor attack surface against 3D object detection through cameras. Due to the dense nature of camera signals, attackers can add 2D triggers with a small size into camera signals, making the attack nearly imperceptible and easy to deploy in practice. Such 2D-oriented backdoor attacks have shown their effectiveness in many 2D object detection tasks Chan et al. [2022], Luo et al. [2023]. Nevertheless, realizing 2D-oriented backdoor attacks against 3D object detection is non-trivial. As illustrated in Figure 1, state-of-the-art LiDAR and camera fusion systems first transform camera signals to align with 2D LiDAR projection, which are then fused with 3D LiDAR features to make detection decisions in a 3D space. Although the transformation of camera signals bridges the gap between 2D and 3D feature spaces, it compromises the association with the injected 2D triggers to the target output. Due to the sparsity of LiDAR points, the resulting transformed camera features are also sparse, causing a limited number of trigger pixels to be observed effectively, thereby substantially diminishing the impact of 2D triggers in 3D object detection. Moreover, due to the dynamicity of LiDAR signals, the applicable trigger pixels after 2D to 3D transformation may not remain consistent across different training samples, further weakening the association between the 2D trigger and target labels. In view of these, it is critical to delve into the potential threats posed by 2D camera-oriented backdoor attacks in influencing 3D object detection.

In this paper, we identify a novel 2D-oriented backdoor attack against the multi-modal 3D object detection system, named BadFusion. BadFusion aims to insert backdoors into the camera and LiDAR fusion-based 3D object detector by only compromising camera inputs with 2D triggers. To obtain effective 2D triggers against the modality transformation in the fusion system, BadFusion develops fusion-aware 2D triggers, which preserve the density of individual triggers while maintaining the trigger pattern consistency across different camera signals. Moreover, considering the inaccessibility of LiDAR signals that are synchronized with camera signals during inference, *i.e.* when deploying the designed triggers to fool the backdoored detector, BadFusion develops LiDAR-free attack approaches, which predicts the 2D LiDAR projection based on camera signals. To the best of our knowledge, this is the first effort in examining 2D-oriented backdoor attacks against fusion-based 3D object detection. We intend to raise community awareness of new backdoor threats in emerging multi-modal fusion systems. Our contributions to this paper are summarized below:

1. We investigate the existing 2D-oriented backdoor attacks against LiDAR and camera fusion systems for 3D object detection. Our research indicates that the fusion system offers effective protection, weakening existing attacks.

2. We propose a new 2D-oriented backdoor attack, named BadFusion, which effectively preserves the 2D backdoor patterns throughout the fusion process and eventually manipulates the 3D predictions.

3. We consider the unavailability of synchronous LiDAR signals when compromising the camera inputs, where a LiDAR-free attack approach is developed to generate LiDAR projection based on camera observations.

4. We extensively evaluate BadFusion against state-of-the-art LiDAR-camera fusion methods with two goals: resizing the bounding boxes and disappearing the objects. BadFusion successfully achieves the two attack goals and outperforms existing 2D-oriented backdoor attacks with a much higher Attack Success Rate (ASR).

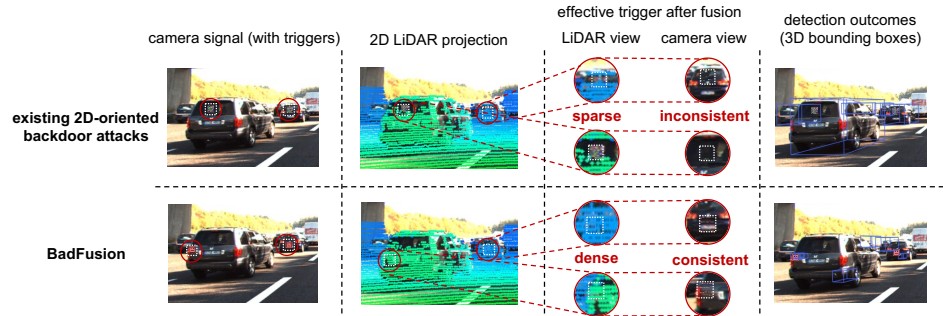

Figure 2: Comparison between existing 2D-oriented backdoor attacks and the proposed BadFusion.

## 2 2D-Oriented Backdoor Attacks

This paper explores the potential of 2D-oriented backdoor attacks in influencing the fusion-based multi-modal 3D perception. This section first introduces the existing backdoor attacks for 2D object detection and then presents the fusion-based 3D object detection systems that involve both 2D and 3D inputs. Our threat model is finally elaborated.

### 2.1 Backdoor Attacks for 2D Object Detection

The mainstream backdoor attack research for object detection centers on 2D perception. An attacker's goal is to use predefined 2D triggers to mislead the target model's predictions. During training, the attacker first poisons $n$ samples of the training dataset $\mathcal{D}_{train} = \{\boldsymbol{x}_i, \boldsymbol{y}_i\}_{i=1}^N$, where $N$ is the number of all training samples, $n \ll N$. Specifically, for a clean sample $(\boldsymbol{x}, \boldsymbol{y})$, the poisoned input $\boldsymbol{x}'$ can be given by

$$\boldsymbol{x}' = \boldsymbol{tr} \odot \boldsymbol{m} + \boldsymbol{x} \odot (1 - \boldsymbol{m}), \tag{1}$$

where $\boldsymbol{tr}$ is the injected trigger; $\boldsymbol{m}$ is a binary mask, using $1$ to represent the location of the trigger and $0$ everywhere else; $\odot$ denotes the element-wise product. Meanwhile, target label $\boldsymbol{y}'$ (different from the original label $\boldsymbol{y}$) is associated with the poisoned input $\boldsymbol{x}'$. The poisoned samples consist of the backdoor dataset $\mathcal{D}_{back}$, which is mixed with rest of clean data $\mathcal{D}_{clean}$ to train the target model $f$. This produces a backdoored model, which misclassifies any poisoned input to the target label while not affecting the prediction of clean samples. The backdoor attack objective is formulated as

$$\min \sum_{(\boldsymbol{x}', \boldsymbol{y}') \in \mathcal{D}_{back}} \mathcal{L}\left(f(\boldsymbol{x}'), \boldsymbol{y}'\right) + \sum_{(\boldsymbol{x}, \boldsymbol{y}) \in \mathcal{D}_{clean}} \mathcal{L}\left(f(\boldsymbol{x}), \boldsymbol{y}\right), \tag{2}$$

where the first and second terms calculate the loss for poisoned and clean samples, respectively. The above problem considers a single modality object detection, which modifies the 2D inputs to mislead 2D predictions, *e.g.*, 2D bounding boxes Chan et al. [2022], Luo et al. [2023]. Instead, this paper targets a multi-modal object detection system with both 2D and 3D inputs for 3D perception, *e.g.*, 3D bounding boxes.

### 2.2 Fusion Pipeline for 3D Object Detection

There are two main categories of research on fusing 2D and 3D inputs for 3D perception. The first projects 3D inputs to a 2D space, which unfortunately often results in severe geometric distortion Chen et al. [2017a], Yang et al. [2018] and thus becomes ineffective for geometric tasks, such as 3D object detection. Therefore, this paper focuses on the second category, which maps 2D inputs to a 3D space for using camera inputs to augment 3D signals. Such fusion has been a promising solution for 3D object detection by preserving critical geometric information Sindagi et al. [2019], Wang et al. [2021], Yin et al. [2021], Li et al. [2022]. As illustrated in Figure 1, one key component of this fusion is the transformation module to map the 2D signal into 3D measurements, which mainly includes three steps. First, the 3D LiDAR signals are projected to a 2D space, such as in the field-of-view (FoV), to derive 2D-LiDAR projection. Then, the 2D camera signals are processed by a camera feature extractor, *e.g.*, 2D CNN, to extract high-level features with semantic information. Finally, the extracted camera features are aligned with 2D LiDAR projection to obtain the camera-based information for each LiDAR point. The transformed camera features will be combined with the LiDAR signals to perform 3D objection detection.

### 2.3 Our Threat Model

This paper focuses on a fusion-based 3D object detection system with both 2D camera and 3D LiDAR inputs. We consider a practical but challenging attack setup: *the objective of the attacker is to launch*

*backdoor attacks for fusion-based 3D object detection by only compromising the camera inputs with 2D triggers*. This attack is more feasible and imperceptible in practice than creating 3D triggers that significantly change the shape and appearance of vehicles. Besides, we consider standard backdoor attack settings: 1) the attacker injects only a small number of poisoned samples into the training dataset; 2) the attacker has no control of the model training process; 3) the attacker has no knowledge about the target model's parameters or architecture.

## 3   The Proposed BadFusion

In order to achieve the aforementioned attack objective, this paper proposes BadFusion, an innovative 2D-oriented backdoor attack against fusion-based 3D object detection. Similar to the attack procedure described in Section 2.1, BadFusion first creates a poisoned dataset. Define the two modality data, 2D camera and 3D LiDAR signals, by $\boldsymbol{x}_c$ and $\boldsymbol{x}_l$, respectively. The poisoned 2D camera data $\boldsymbol{x}_c'$ is created by injecting the 2D trigger $\boldsymbol{tr}$ to $\boldsymbol{x}_c$ based on (1). Meanwhile, the target label $y'$ is associated with the poisoned camera input $\boldsymbol{x}_c'$. After that, the poisoned camera inputs $\boldsymbol{x}_c'$, remaining clean camera inputs $\boldsymbol{x}_c$, and LiDAR inputs $\boldsymbol{x}_l$ are jointly used to train the backdoored fusion model $f$. This optimization problem is formulated as

$$\min \sum_{(\boldsymbol{x}_l, \boldsymbol{x}_c', \boldsymbol{y}') \in \mathcal{D}_{back}} \mathcal{L}\left(f(\boldsymbol{x}_l, \boldsymbol{x}_c'), \boldsymbol{y}'\right) + \sum_{(\boldsymbol{x}_l, \boldsymbol{x}_c, \boldsymbol{y}) \in \mathcal{D}_{clean}} \mathcal{L}\left(f(\boldsymbol{x}_l, \boldsymbol{x}_c), \boldsymbol{y}\right). \tag{3}$$

### 3.1   Design Challenges

Although existing backdoor attacks against single-modality systems, *i.e.*, camera-only inputs, can successfully mislead 2D object detection, BadFusion cannot directly follow their attack procedure. As discussed in Section 2.2, the target fusion model $f$ in (3) needs to transform the poisoned camera signal $\boldsymbol{x}_c'$ from 2D to 3D measurement for data fusion purposes. Unfortunately, this transformation breaks the association with the injected 2D trigger to the target output. Specifically, we identify the following two key challenges: (1) *Trigger sparsity*. Due to the sparsity of 3D LiDAR points, only a few camera pixels are transformed into LiDAR features and subsequently used for object detection. Thus, most pixels of the 2D triggers are ignored in the fusion-based object detection system, making it hard to mislead the prediction of the target model. (2) *Trigger inconsistency*. Due to the dynamicity of LiDAR data, the same LiDAR point may correspond to different pixels of the transformed camera signal. Thus, the effective trigger pixels become inconsistent among inputs after transformation. Consequently, the effective trigger pixels during inference are inconsistent with those during training, weakening the association between trigger patterns and target labels. Figure 2 illustrates the challenges of existing attacks. The triggers injected by existing attacks are both sparse and inconsistent due to the transformation, thus ineffective in misleading the fusion model.

### 3.2   Fusion-Aware 2D Trigger Design

To address these challenges, BadFusion employs the fusion-aware 2D triggers tailored for multi-modal fusion systems. These triggers aim to preserve dense and consistent patterns against the transformation module of the fusion pipeline. To enhance *trigger density*, BadFusion intends to maximize the effective pixels in 2D triggers after transformation. Recall that the 2D camera trigger aligns with the 2D LiDAR projection to extract applicable camera features for multi-modal fusion, as shown in Figure 1. Hence, we propose to identify the dense region of the 2D LiDAR projection for trigger placement, where only contiguous dense regions are considered to make 2D triggers easy to implement in reality. Besides, we identify another challenge from the availability of LiDAR signals. Although the LiDAR signals of training samples are accessible to the attacker, LiDAR signals in the inference phase are usually unavailable. Therefore, we introduce a LiDAR-free method for BadFusion by predicting the dense regions of the 2D LiDAR projection, which is detailed in Section 3.3.

Additionally, to enhance *trigger consistency*, BadFusion intends to maximize the consistent trigger patterns among different inputs. Conventional 2D backdoor attacks optimize triggers with various colors of pixels towards different goals, such as high attack success rate, clean data accuracy, and high stealthiness Liu et al. [2018], Zhao et al. [2020], Zhong et al. [2020], Garg et al. [2020]. The impact of these colorful pixels will be diminished in the fusion system after the 2D to 3D transformation, as the effective pixels of an optimized trigger after the transformation vary among different inputs. Thus, instead of generating complex and imperceptible triggers, we introduce a simple yet effective approach to create 2D triggers with (almost) solid colors for all pixels. These triggers with a solid color remain consistent after transformation across different inputs, thereby largely reducing the discrepancy between different backdoor samples.

## 3.3 LiDAR-Free Attack

In many real-world scenarios, the attacker does not have access to the LiDAR signal that is synchronized with the camera signal, especially during inference, *i.e.*, when deploying the designed 2D trigger to fool the backdoored fusion model. Hence, the absence of LiDAR signals poses challenges to identifying the densely populated regions of the 2D LiDAR projection where the 2D trigger should be implemented. To address this issue, we propose a LiDAR-free BadFusion approach by predicting dense regions of the 2D LiDAR projection based on camera signals. We convert this region prediction task to an object detection task, where the object becomes the densest region in the 2D LiDAR projection. To achieve this, we create a training dataset containing camera signals and the bounding boxes of the densest areas, denoted by $(x, y, w, h)$, where $x$ and $y$ are the center coordinates, and $w$ and $h$ are the width and height of bounding box, respectively. Here, we set $w$ and $h$ the same as the width and height of the injected trigger $\boldsymbol{tr}$. For each vehicle, we annotate a bounding box that contains most points in 2D LiDAR projection. Then, we train a dense region detector $f_{2d-lidar}$ to predict the bounding boxes based on the Faster R-CNN framework Ren et al. [2015] with a VGG backbone. Our evaluation results show that the detector can successfully identify the dense areas and facilitate the backdoor attacks even without knowing the LiDAR signals, which achieves comparable performance to the LiDAR-aware attack.

## 3.4 Overall Algorithm Design

Algorithm 1 outlines the overall procedure of BadFusion as shown in Appendix A.

# 4 Evaluation

In this section, we first detail our experimental framework (dataset, implementation & training details, evaluation metrics) and then present the evaluation results of the proposed BadFusion. We further demonstrate the effectiveness of BadFusion against mainly Point-line Camera-to-LiDAR fusion methods in 3D object detection and also benchmark our approach against three state-of-the-art backdoor detection methods. Lastly, we conduct an ablation study to elucidate the internal mechanics of the BadFusion.

## 4.1 Evaluation Settings

**Dataset.** We use the KITTI dataset Geiger et al. [2013] in the evaluation. The dataset collects real traffic environments from Europe Street for 3D detection tasks, comprising $7,481$ labeled training frames and $7,518$ unlabeled test samples. Additional details are presented in Appendix C.

**LiDAR-camera fusion methods.** In this work, we evaluate backdoor attacks against widely used LiDAR-camera fusion methods. In the main paper, we report the evaluation results on MVX-Net, a single-stage fusion model Sindagi et al. [2019]. The results for other LiDAR-camera fusion methods are presented in Appendix. To train the fusion model, we adopt common data augmentation techniques, including resizing, rotation, scaling, translation, and flip[1]. We use FocalLoss Lin et al. [2017] for classification and SmoothL1Loss Huber [1992] for bounding box regression, respectively. The fusion models are trained using an AdamW optimizer with a learning rate of $0.002$ and a weight decay parameter of $0.01$ for $70$ epochs.

**Attack goals.** To manipulate the prediction of vehicles (*Car* class in the KITTI dataset), we consider two attack goals. 1) Resizing attack: the attacker aims to reduce the sizes of target bounding boxes to mislead the prediction as a smaller vehicle, 2) Disappear attack: the attacker aims to make the vehicle disappear from detection. Figure 3 illustrates an example of the two goals of attacks. Additional details on Attack goals are presented in Appendix C.1. In this section, we report most evaluation results based on the resizing attacks. The effectiveness of the disappear attacks is presented in the ablation study (Section 4.3).

**Baseline attacks and attack setup.** We compare the proposed BadFusion with three state-of-art 2D-oriented backdoor attacks, including OptimizedTrigger Liu et al. [2018], BadDet Chan et al. [2022] and UntarOD Luo et al. [2023]. To make a fair comparison, for all attacks, we poison 15% training data using a trigger with the size of $15 \times 15$ and maintain consistent training or experimental settings. Additional details on Baseline attacks and their attack setup are presented in Appendix C.1.

**Evaluation metrics.** We evaluate the effectiveness of the backdoor attacks based on three well-established metrics: Clean data mAP, Attack Success Rate (ASR), and Poisoned data mAP. An

---

[1]Data augmentation techniques are implemented by Resize, GlobalRotScaleTrans, RandomFlip3D using mmdetection3d: https://github.com/open-mmlab/mmdetection3d

Table 1: Comparison between existing backdoor attacks and proposed BadFusion against MVX-Net. We perform resizing attacks to reduce the bounding boxes of predicted vehicles. Clean model shows the performance of the fusion model without backdoor attacks.

| Backdoor attack | Clean data mAP (%) ↑ | Poisoned data mAP (%) ↓ | ASR (%) ↑ |
|---|---|---|---|
| Clean model | 93.75 | - | - |
| OptimizedTrigger | 18.90 | 21.12 | 49.49 |
| BadDet | 36.14 | 48.21 | 39.32 |
| UntarOD | 64.98 | 37.57 | 46.74 |
| (LiDAR-aware) BadFusion | 88.65 | 1.61 | 96.74 |
| BadFusion | 88.65 | 3.05 | 95.28 |

effective backdoor attack should achieve high clean data mAP, high ASR, and low poisoned data mAP. Additional details on the definitions of the Evaluation metrics are presented in Appendix C.1.

## 4.2 Main Evaluation Results

Table 1 compares our proposed BadFusion attack with existing backdoor attacks. The results show that the existing backdoor attacks (OptimizedTrigger, BadDet, UntarOD) are ineffective in manipulating the fusion detector's predictions. All the attacks result in less than 50% ASR and relatively high poisoned data mAP. This is mainly due to the sparse and inconsistent trigger patterns during the fusion process as discussed in Section 3.1. Our proposed BadFusion attack addresses the problem and successfully performs backdoor attacks achieving over 95% ASR and around 3% poisoned data mAP. In the meanwhile, BadFusion can still provide accurate predictions on the clean samples without triggers and achieves much higher clean data mAP compared with the baseline attacks. Additionally, in BadFusion, we assume the attacker has no information about LiDAR signals and trains a model to predict the dense LiDAR region. To investigate the effectiveness of the dense region detector, we compare BadFusion with a LiDAR-ware version of BadFusion, where we assume LiDAR signals are accessible and the dense region can be directly calculated. We find that although LiDAR-ware BadFusion achieves a better attack performance. However, the gap between BadFusion and LiDAR-ware BadFusion is marginal, which suggests the effectiveness of the dense region detector.

## 4.3 Ablation Study

In this section, we conduct ablation studies to demonstrate the effectiveness of BadFusion with different attack goals and trigger patterns. Additional experiments are presented in Appendix C.5.

**Effectiveness of BadFusion with different attack goals**. We first investigate the attack performance with two goals in BadFusion: resizing bounding boxes and disappearing objects. In disappearing attack, we use two poisoning strategies: moving the center coordinates of bounding boxes farther or closer in the poisoned data. As shown in Appendix Table 2, all the attacks achieve good performance with high ASR and low poisoned data mAP. Additionally, we find that, compared with disappear attack (farther), moving bounding boxes closer is more effective for disappearing attacks.

**Effectiveness of BadFusion with different trigger patterns.** In the evaluation, we consider two trigger patterns. 1) Solid pattern: using a solid color for all pixels in the trigger and 2) almost solid pattern: using a solid color for most pixels while only a few pixels are applied with other colors. The almost solid pattern applies to many real-world scenarios, *e.g.*, emojis or decals used in vehicle stickers, which makes the trigger more stealthy. Figure 6 in the Appendix shows the two patterns used in the evaluation. As shown in Appendix Table 9, using almost solid patter, BadFusion can still achieve an ASR of 79.51%. This indicates the severe security risks of BadFusion in the real world.

## 5 Conclusion

This paper presents the first analysis of 2D-oriented backdoor attacks against LiDAR-camera fusion for 3D object detection. By analyzing the existing 2D-oriented backdoor attacks, we find that these attacks are ineffective against fusion models due to the sparsity and inconsistency of backdoor triggers introduced during the fusion process. To address these challenges, we propose BadFusion, an innovative fusion-aware backdoor attack against 3D object detection. By maximizing both effective trigger pixels and consistent trigger patterns among different inputs, BadFusion successfully performs backdoor attacks against state-of-the-art LiDAR-camera fusion methods and realizes two attack goals: resizing the bounding boxes and disappearing the objects. Compared with existing 2D-oriented backdoor attacks, BadFusion achieves a much higher attack success rate and low Poisoned data mAP. We hope our analysis will enhance safety awareness for autonomous driving and promote further research in this field.

## Acknowledgement

This work is supported in part by the National Science Foundation under Grants CCF-2106754, CCF-2221741, and CNS-2151238.

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

# Appendix

## A   Algorithm

The algorithm outline of the proposed BadFusion is listed as Algorithm 1. In the training phase, the attacker first injects fusion-aware 2D triggers into $n$ samples of camera training inputs. These triggers are placed at the densest region of the corresponding 2D LiDAR projection with a solid color. Next, both the clean and backdoor datasets are used to train the target fusion-based 3D object detector $f$. Once the training is complete, to mislead the target model in the inference phase, the attacker attaches the trigger to vehicles based on the position predicted by the dense region detector $f_{2d-lidar}$. Eventually, BadFusion will mislead the prediction of the target detection $f$ to predict vehicles with the designed trigger.

---

**Algorithm 1** Algorithm Procedure of BadFusion

---

**Input**: target 3D object detector $f$, trigger $\boldsymbol{tr}$ (designed with uniform color), training dataset $\mathcal{D}_{train}$, number of backdoor training samples $n$, dense area detector of 2D LiDAR projection $f_{2d-lidar}$.

1: *// Training Phase*
2: Sample $n$ training samples from $\mathcal{D}_{train}$ for attacks. Rest clean training samples are denoted as a clean dataset $\mathcal{D}_{clean}$
3: Initialize a backdoor dataset $\mathcal{D}_{back} = \emptyset$
4: **for** $i = 1$ to $n$ **do**
5:     *// Inject trigger to training data $(\boldsymbol{x}_{lidar}, \boldsymbol{x}_{camera}, y)$*
6:     Calculate 2D LiDAR projection of $\boldsymbol{x}_{lidar}$ and get the bounding box $(x, y, w, h)$ with most projection points for each vehicle
7:     Add the trigger $\boldsymbol{tr}$ to the bounding box in the camera signal $\boldsymbol{x}_{camera}$ for each vehicle: $\boldsymbol{x}_{camera}[x - \frac{w}{2} : x + \frac{w}{2}, y - \frac{h}{2} : y + \frac{h}{2}] \leftarrow \boldsymbol{tr}$. The poisoned camera signal is denoted as $\mathcal{A}(\boldsymbol{x}_{camera}, \boldsymbol{tr})$
8:     Change the label to the target label $y'$
9:     Add the backdoor data to the backdoor dataset $\mathcal{D}_{back} \leftarrow \mathcal{D}_{back} \cup (\boldsymbol{x}_{lidar}, \mathcal{A}(\boldsymbol{x}_{camera}, \boldsymbol{tr}), y')$
10: **end for**
11: Train the fusion detector $f$ on both the clean dataset $\mathcal{D}_{clean}$ and the backdoor dataset $\mathcal{D}_{back}$.

12: *// Inference Phase*
13: Predict bounding box of the most dense region $(x, y, w, h)$ using $f_{2d-lidar}$ and attach the trigger for attacks.

---

## B   Related Works

### B.1   Backdoor Attacks

Backdoor attacks aim to inject malicious behavior into a target model and change the model's prediction for the input samples with the trigger pattern. One of the earliest backdoor attacks, called BadNets Gu et al. [2017], was introduced by Gu et al. This attack injected a simple image trigger pattern into the training dataset, causing the model to produce misleading predictions for samples containing the trigger pattern. Subsequent research has advanced backdoor attacks with different objectives, such as stealthy attacks with invisible triggers Chen et al. [2017b], Li et al. [2021b], attacks without manipulating labels (clean label attack) Turner et al. [2018], and attacks that are resistant to transfer learning Yao et al. [2019], Wang et al. [2020]. Most backdoor attacks focus on image tasks, such as image classification Gu et al. [2017] and 2D object detection Chan et al. [2022], Luo et al. [2023], that involve 2D triggers. Recently, it has been discovered that backdoor attacks can also manipulate 3D detection prediction Li et al. [2021a], Xiang et al. [2021], Zhang et al. [2022]. However, these attacks rely on 3D LiDAR triggers, which are easily detectable and impractical to implement in real-world scenarios. In our work, we present a novel 2D-oriented backdoor attack that injects 2D triggers in the training data while aiming to manipulate 3D prediction.

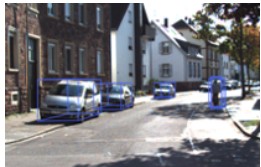 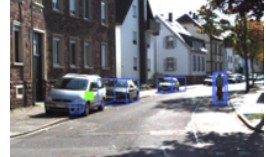 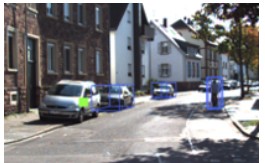

(a) Clean model.      (b) Resizing attack.      (c) Disappear attack.

Figure 3: Examples of different attack goals in BadFusion. Fig (a) shows the predictions of a clean model without backdoor triggers. Fig (b) shows the predictions of a resizing attack, where the attack reduces the size of the predicted bounding box. Fig (c) shows the predictions of a disappear attack, where the attack removes the predicted bounding box of a vehicle from the prediction for disappearing the vehicle.

Table 2: Performance of BadFusion with different attack goals. BadFusion is effective in both resizing the bounding box prediction and disappearing the objects.

| Attack goal | Clean data mAP (%) ↑ | Poisoned data mAP (%) ↓ | ASR (%) ↑ |
|---|---|---|---|
| Resizing | 88.65 | 3.05 | 95.28 |
| Disappear (farther) | 87.35 | 6.95 | 89.93 |
| Disappear (closer) | 92.86 | 19.03 | 94.74 |

## B.2 LiDAR-camera Fusion for 3D Object Detection.

LiDAR-camera fusion has emerged as a promising solution for 3D object detection. By combining complementary signals, the fusion model achieves state-of-the-art detection performance. One of the key challenges of fusing LiDAR and camera signals is how to align these two signals in the same measurement. Given the advantages of spatial information provided by LiDAR sensors, recent work mainly focuses on aligning the camera features to LiDAR Sindagi et al. [2019], Wang et al. [2021], Yin et al. [2021], Li et al. [2022], Chen et al. [2022]. For example, Sindagi et al. proposed MVX-Net Sindagi et al. [2019] that first projects LiDAR points onto the image and then appends the camera features to LiDAR points with the same location index. Chen et al. Chen et al. [2022] leveraged the similar fusion method, by adding camera features to the important LiDAR features only. The importance of LiDAR points is determined by their proposed Focals Conv operation. These fusion methods inherently provide a strong defense against backdoor attacks, since the backdoor triggers injected into camera signals become ineffective after the alignment in the fusion methods. However, in our work, we reveal the vulnerability of LiDAR-camera fusion using the proposed BadFusion attack.

# C    Evaluation.

In this section, we provide additional details on the dataset, attack goals, Baseline attacks, and evaluation metrics.

## C.1   Experimental Setup.

**Dataset.** In our experiments on the KITTI dataset Geiger et al. [2013], since the ground-truth of the test data is unavailable, we split the training data into a train set and a validation set with $3,712$ and $3,769$ samples, respectively, following the train/valid split process in previous work Chen et al. [2016]. To conduct data poisoning on the train set, we select cars categorized under easy and medium difficulty from the KITTI dataset. For evaluation, we focused on cars labeled as easy difficulty in the validation set, which can be accurately predicted by the clean model.

**Attack goals.** As specified in the main text, we consider two attack goals. 1) Resizing attack, 2) Disappear attack. These attacks pose a significant threat to autonomous driving systems. Note that the existing work achieves disappear attacks by removing bounding boxes from the labels. However, removing bounding boxes is ineffective for optimizing the poisoned model, as the empty bounding boxes are not presented in the labels and are not optimized in the optimization objective (Eq. 3). To

achieve disappear attacks, we relocate the center coordinates of the bounding boxes in the poisoned data and make them closer or farther from the target vehicle, denoted as disappear attack (closer) and disappear attack (farther). We use the relocated bounding boxes as the label in the poisoned training data. The relocation breaks the connection between input signals and the bounding box labels. We find that our proposed two attack methods can effectively remove the bounding box prediction, making the vehicle disappear from predictions.

**Baseline attacks and attack setup.** BadDetChan et al. [2022] and UntarOD Luo et al. [2023] are targeted against 2D object detection tasks and OptimizedTrigger Liu et al. [2018] is a general backdoor attack with optimized triggers. We implement UntarOD based on their open-source code[2] and implement OptimizedTrigger and BadDet following their papers. The OptimizedTrigger attack, originally designed for the image classification problem, has been adapted to align with the object detection settings of the MVX-Net model. In this case, we first optimized the trigger and performed poisoning accordingly.

**Evaluation metrics.** The additional information on three metrics used in our work are as follows: First, *Clean data mAP* refers to the mean average precision calculated on clean samples without backdoor triggers when predicted by the poisoned model. Typically, an attacker's goal is to design a poisoned model that performs well on benign samples, *i.e.*, achieving a high Clean data mAP. Second, *Attack Success Rate (ASR)* represents the proportion of attacked samples that successfully achieve the backdoor objective based on different types of attacks when influenced by the poisoned model. Specifically, for resizing attacks, we define ASR as the ratio of bounding box sizes decreased when a trigger is applied. For disappear attacks, we define ASR as the ratio of the bounding box disappearing when a trigger is applied. Third, *poisoned data mAP*, refers to the mean average precision calculated on poisoned samples when predicted by the poisoned model. An effective backdoor attack should achieve high clean data mAP, high ASR, and low poisoned data mAP.

## C.2 BadFusion Setup.

In the proposed BadFusion attack, we determine the $(x, y)$ coordinates for trigger insertion based on dense regions in the corresponding 2D LiDAR projections onto image planes. We filter out LiDAR points located within the bounding box of the vehicle intended for poisoning and employ a sliding window to identify the densest region with most LiDAR points. For resizing attacks, we reduce the sizes of the 3D Bounding Box by 75% to facilitate a reduced size bounding box attack. For disappearing attacks, we change the (x, z) coordinate by doubling or halving the values of x and z, *i.e.*, making the vehicles farther or closer distance from the target vehicle.

## C.3 Fusion Methods

**MVX-Net.** We trained the MVX-Net fusion model Sindagi et al. [2019] using Faster R-CNN as backbone. This model integrates a frozen 50-layer ResNet architecture with a Feature Pyramid Network neck in its image backbone to enhance feature representation. Point cloud processing is conducted using a voxelization block, and the Camera and LiDAR features are fused using the PointFusion method. Finally, the fused features are processed with the Dynamic Voxel Feature Encoding module and 3D Region Proposal Network for 3D Object Detection.

**Focals Conv-F.** We trained the Focals Conv-F fusion model Chen et al. [2022] utilizing the PVRCNN Shi et al. [2020] as the base model. The Focals Conv-F fusion model employs the BaseBEVBackbone as the 2D backbone and their proposed Focal Sparse Convolutional Networks for processing the 3D backbone, leveraging image data and BeV mappings. The extracted Camera and LiDAR features undergo point-wise fusion and serve as input for the proposed Focal Sparse Convolutional network. This network determines which input features deserve dilation and adjusts the output shapes dynamically, based on the predicted cubic importance. The resulting features are processed by a set of fully connected layers to predict the 3D bounding boxes of the objects in the scene.

**EPNet.** We trained the EPNet F fusion model Huang et al. [2020] utilizing the two-stream Region Proposal Network (RPN) for proposal generation and a refinement network for bounding box refining. In EPNet, the two-stream RPN is composed of a geometric stream and an image stream, which produce the point features and semantic image features, respectively. Finally, the LI-Fusion module

---

[2]https://github.com/Chengxiao-Luo/Untargeted-Backdoor-Attack-against-Object-Detection

Table 3: Comparison between existing backdoor attacks and proposed BadFusion against Focals Conv-F. We perform resizing attacks to reduce the bounding boxes of predicted vehicles. The BadFusion attacks achieve comparable results with LiDAR-aware BadFusion attacks, where the LiDAR information is accessible to the attacker.

| Backdoor attack | Clean data mAP (%) ↑ | Poisoned data mAP (%) ↓ | ASR (%) ↑ |
|---|---|---|---|
| Clean model | 94.83 | - | - |
| OptimizedTrigger | 94.74 | 95.02 | 5.45 |
| BadDet | 96.36 | 94.56 | 6.91 |
| UntarOD | 93.76 | 93.23 | 7.89 |
| (LiDAR-aware) BadFusion | 95.13 | 23.11 | 91.72 |
| BadFusion | 95.13 | 28.00 | 90.54 |

Table 4: Comparison between existing backdoor attacks and proposed BadFusion against EPNet. We perform resizing attacks to reduce the bounding boxes of predicted vehicles. The BadFusion attacks achieve comparable results with LiDAR-aware BadFusion attacks, where the LiDAR information is accessible to the attacker.

| Backdoor attack | Clean data mAP (%) ↑ | Poisoned data mAP (%) ↓ | ASR (%) ↑ |
|---|---|---|---|
| Clean model | 94.38 | - | - |
| OptimizedTrigger | 95.41 | 6.87 | 95.45 |
| BadDet | 94.54 | 93.76 | 5.93 |
| UntarOD | 95.26 | 12.90 | 92.03 |
| (LiDAR-aware) BadFusion | 95.65 | 6.45 | 94.30 |
| BadFusion | 95.65 | 8.30 | 92.44 |

establishes the fine-grained point-wise correspondence between LiDAR and camera image data, and fuses the point features and semantic image features based on the correspondence generated by the grid generator. The LI-Fusion module adaptively estimates the importance of the image semantic features and fuses them with the point features to enhance the 3D object detection performance.

## C.4   Evaluation Results for Focals Conv-F Fusion and EPNet Fusion

We perform 2D-oriented backdoor attacks against Focals Conv-F and EPNet fusion models. Following the main text, we set a trigger size of $15 \times 15$ and a poisoning rate of $15\%$ to poison the model. We compare our proposed BadFusion attack with existing backdoor attacks on the Focals Conv-F fusion model in Table 3 and EPNet fusion model in Table 4. As shown in Table 3, the existing backdoor attacks (OptimizedTrigger, BadDet, UntarOD) result in a very low ASR and unchanged Poisoned mAP, despite performing poisoning, which shows the existing backdoor attacks fail to mislead the fusion models. On the other hand, the proposed BadFusion can effectively affect the detection performance of the Focals Conv-F fusion method, achieving an ASR of $90.54\%$. Additionally, the difference between LiDAR-free BadFusion and LiDAR-aware BadFusion is marginal. The clean mAP in Table 3 for the existing backdoor attacks is higher as compared to clean mAP in Table 1 because the existing backdoor attacks for MVX-Net learned memorization instead of learning poisoned behavior (*i.e.* irrespective of whether trigger is present or not they predicted resized (smaller) bounding box), thereby affecting clean mAP. In Focals Conv-F fusion method, the poisoning does not have any impact on the existing backdoor attacks. So, they behave similar to clean model and have higher clean mAP.

As shown in Table 4, the existing backdoor attack (BadDet) result is consistent with the existing backdoor attacks results for Focals Conv-F fusion, where they have a low ASR and unchanged Poisoned mAP. Whereas, the existing backdoor attack (OptimizedTrigger, UntarOD) learned the backdoor behavior learning, as the EPNet fusion method adds continuous image feature segmentation instead of sparse image features during fusion, unlike MVX-Net or Focals Conv-F, resulting in weaker built-in defense against backdoor attacks. The proposed BadFusion can also effectively affect

Table 5: Impact of different trigger sizes in the Inference for the proposed BadFusion against MVX-Net under resizing attack, disappear attack (farther), and disappear attack (closer).

| Attack Goal | Inference Trigger Size | Clean data mAP (%) ↑ | Poisoned data mAP (%) ↓ | ASR (%) ↑ |
|---|---|---|---|---|
| | 10x10 | | 14.00 | 90.06 |
| | 20x20 | | 1.17 | 96.74 |
| Resizing Attack | 30x30 | 93.75 | 0.30 | 97.38 |
| | 40x40 | | 0.16 | 97.76 |
| | 50x40 | | 0.14 | 97.72 |
| | 10x10 | | 22.29 | 92.61 |
| | 20x20 | | 2.81 | 89.32 |
| Disappear Attack (farther) | 30x30 | 93.75 | 0.67 | 88.57 |
| | 40x40 | | 0.19 | 88.03 |
| | 50x40 | | 0.15 | 87.38 |
| | 10x10 | | 44.53 | 95.96 |
| | 20x20 | | 11.37 | 93.89 |
| Disappear Attack (closer) | 30x30 | 93.75 | 3.39 | 92.91 |
| | 40x40 | | 1.57 | 92.40 |
| | 50x40 | | 0.77 | 91.42 |

the detection performance of the EPNet fusion method, achieving an ASR of $94.30\%$ with better Poisoned data mAP and Clean data mAP as compared to existing backdoor attacks.

### C.5 Additional Ablation Study

**Different trigger sizes during inference.** In previous backdoor attacks, the attacker uses the same trigger size in the training and inference phase to keep the same trigger pattern. In this experiment, we use trigger sizes of $15 \times 15$ in the training and different trigger sizes (from $10 \times 10$ to $50 \times 50$) in the inference phase. Since we use the solid color in the trigger, the increased trigger size does not affect the consistency of the trigger pattern, but can potentially increase the number of effective trigger pixels after fusion. Table 5 shows the results of BadFusion with resizing attack goal. For resizing attacks, increasing trigger sizes in the inference significantly improves the attack performance of BadFusion. For example, with the trigger size of $30 \times 30$, the ASR of BadFusion reaches over $97\%$. This is mainly due to the increased number of effective trigger pixels on 2D LiDAR projection.

**Selection of poisoned samples.** We investigate the impact of poisoned sample selection on the attack performance. We found that the number of effective trigger pixels after fusion plays an important role in backdoor attack performance. In the main paper, we select poisoned samples whose effective trigger pixels follow Gaussian distributions. In this experiment, we select two additional distributions for poison sample selection: left-skewed distribution, and right-skewed distribution, as shown in Figure 6. The other attack setups follow the main paper: we use a trigger size of $15 \times 15$ and a poisoning rate of $15\%$.

We find that the selection of poisoned samples significantly affects the attack performance. As shown in Table 6, the normal distribution shows a much better attack performance than other selections (low Poisoned mAP and high ASR). Any deviation (left-skewed distribution, or right-skewed distribution) in the consistency or uniformity of transformed trigger feature pixels across training samples hampers the poisoned model's ability to learn the targeted poisoned behavior. The performance of the poisoned model drops to $87.01\%$ for left-skewed distribution and $60.57\%$ for right-skewed distribution. We believe this is due to that the normal distribution selection better covers different trigger patterns, *i.e.*, different numbers of effective trigger pixels, that may occur in the inference phase. Therefore, the injected trigger can be better "generalized" to attacks in the inference.

**Experiment details on the impact of trigger size and poisoning rate.** We investigate the impact of poisoning rate and trigger size. In most of experiments, we set poisoning rate as $15\%$ and trigger size as $15 \times 15$. Here we consider additional poisoning rate of $20\%$ and trigger size as $20 \times 20$. We report the results in Table 7. We find that increasing trigger size and poisoning rate is not necessary for improving attack performance. For example, a $20$ poisoning rate proves optimal for a poisoned model with a $20 \times 20$ trigger size when compared to their respective counterparts with $20\%$ and $15\%$ poisoning rates. However, a $15\%$ poisoning rate is more suitable for a poisoned model with a $15 \times 15$ trigger size. We think this is mainly due to that increasing the trigger size and poisoning rate may

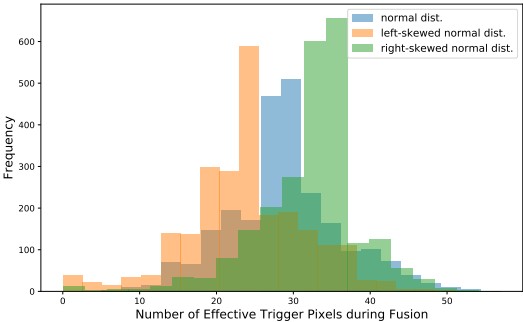

Figure 4: Different distributions of backdoor sample selection in BadFusion against MVX-Net.

Table 6: Impact of different poison sample selections in BadFusion.

| Sample selection | Clean mAP (%) ↑ | Poisoned mAP (%) ↓ | ASR (%) ↑ |
|---|---|---|---|
| normal dist. | 88.65 | 3.05 | 95.28 |
| left-skewed dist. | 91.88 | 19.44 | 87.01 |
| right-skewed dist. | 40.24 | 42.36 | 60.57 |

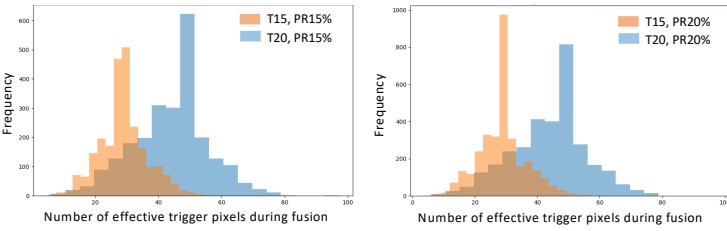

Figure 5: Distribution of effective trigger pixels during fusion using different trigger sizes. We conduct BadFusion attacks against the MVX-Net fusion model for resizing attacks. T15 and T20 represents trigger size $15 \times 15$ and $20 \times 20$, respectively. PR15% and PR20% represents poisoning rate 15% and 20%, respectively.

also increase the inconsistency of trigger patterns among poisoned data, which further amplifies the challenge of backdoor attacks discussed in Section 3.1.

Figure 5 shows the distributions of effective trigger pixels of poisoned samples during fusion. As shown in Figure 5, with the same poisoning rate, the small trigger size ($15 \times 15$) results in a small range of effective trigger pixels, varying from 0 to 60, predominantly ranging from 25 to 35, while the large trigger size ($20 \times 20$) results in a much larger range of effective trigger pixels. The small range of effective trigger pixels makes the trigger pattern more consistent between samples in the training and inference phase, leading to a better attack performance for a small trigger size.

Additionally, the BadFusion poisoned model with poisoning rate 15% and $15 \times 15$ trigger size, outperforms the poisoning rate 20% poisoned model. We think this is because the increment of 5% data (shifting from 15% poisoning rate to 20% poisoning rate) only contributes an additional 342 samples within the consistent number of effective trigger pixels range, while introducing 458 samples within the inconsistent number of effective trigger pixels range. This imbalance disrupts the model's overall learning process. Conversely, the $20 \times 20$ trigger size and poisoning rate 20% poisoned model surpasses the 15% poisoning rate model. Here, the additional 5% data (shifting from 15% poisoning rate to 20% poisoning rate) contributes 443 samples within the consistent number of effective trigger pixels range and only 357 within the inconsistent number of effective trigger pixels range, thereby facilitating the model's learning process.

**Impact of switching to an almost solid trigger pattern on baselines.**

Table 7: Performance of BadFusion using different poisoning rates and trigger sizes.

| Trigger size | Poisoning rate (%) | Clean data mAP (%) ↑ | Poisoned data mAP (%) ↓ | ASR (%) ↑ |
|---|---|---|---|---|
| 15x15 | 15 | 88.65 | 3.05 | 95.28 |
| 15x15 | 20 | 84.25 | 5.69 | 91.52 |
| 20x20 | 15 | 93.17 | 45.34 | 62.44 |
| 20x20 | 20 | 89.09 | 47.23 | 84.03 |

Table 8: Impact of switching to emoji trigger on baselines on MVX-Net Fusion Method.

| Backdoor attack | Clean data mAP (%) ↑ | Poisoned data mAP (%) ↓ | ASR (%) ↑ |
|---|---|---|---|
| OptimizedTrigger | 34.95 | 47.23 | 37.35 |
| BadDet | 29.99 | 41.49 | 45.08 |
| UntarOD | 30.98 | 42.89 | 40.06 |

Table 9: Performance of BadFusion using different trigger patterns.

| Trigger pattern | Clean data mAP (%) ↑ | Poisoned data mAP (%) ↓ | ASR (%) ↑ |
|---|---|---|---|
| Solid | 88.65 | 3.05 | 95.28 |
| Almost solid | 90.12 | 27.83 | 79.51 |

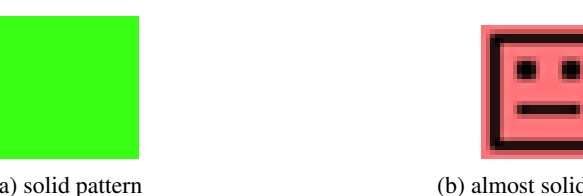

(a) solid pattern          (b) almost solid pattern

Figure 6: Different trigger patterns used in BadFusion.

Table 8 presents the impact of switching to an almost solid trigger pattern on MVX-Net Fusion Method in comparison to the baselines (existing backdoor attacks). The results align with the main findings presented in Table 1, where the existing backdoor attacks (OptimizedTrigger, BadDet, UntarOD) prove ineffective in manipulating the fusion detector's predictions, even when employing a almost solid trigger pattern for the backdoor attack. Conversely, the effectiveness of BadFusion with a more stealthy almost solid trigger pattern is demonstrated in Table 9.

## C.6  Comparison between LiDAR-aware and LiDAR-Free BadFusion attacks

Table 1 and 3 show that LiDAR-free BadFusion achieves comparable performance with LiDAR-aware BadFusion, where we assume the attack has access to the LiDAR signals in the inference phase. Here, we present the example triggers injected by LiDAR-aware and LiDAR-Free BadFusion attacks in Figure 7. Even though the LiDAR-Free approach does not position the trigger at the vehicle's densest regions, the chosen trigger locations are still effective for backdoor attacks. This suggests the effectiveness of the dense region detector in BadFusion.

## C.7  Examples of BadFusion with different attack goals

In this section, we present examples of BadFusion attacks achieving different attack goals. Figure 8 illustrates the detection results of the Clean Model and poisoned model under resizing attacks and disappearing attacks. These comparisons demonstrate the effectiveness of the proposed BadFusion across different attack goals.

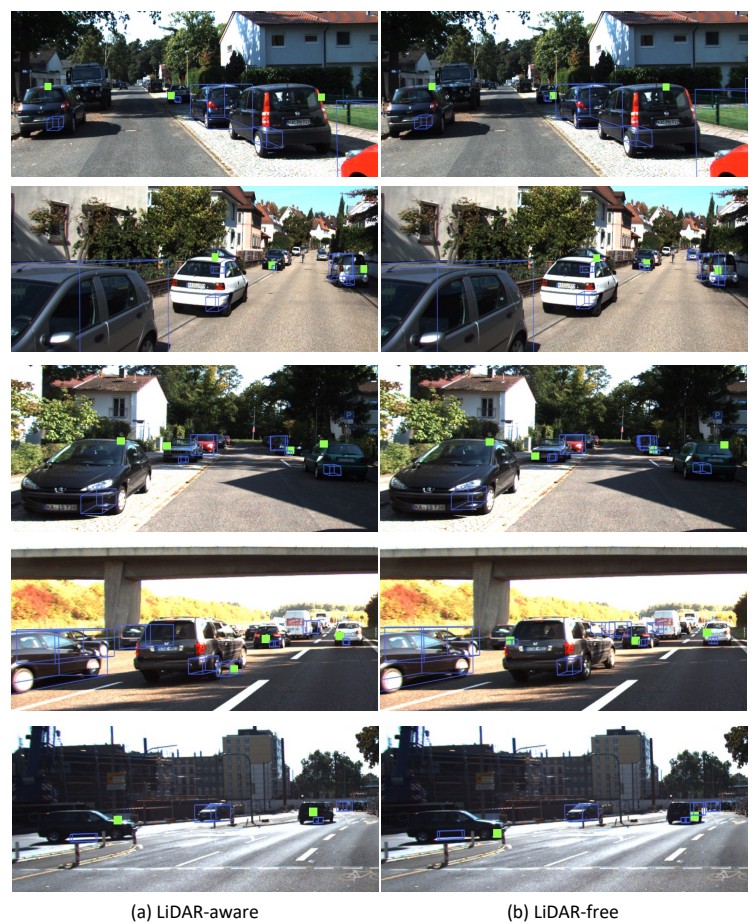

(a) LiDAR-aware        (b) LiDAR-free

Figure 7: Comparison between LiDAR-aware and LiDAR-free BadFusion.

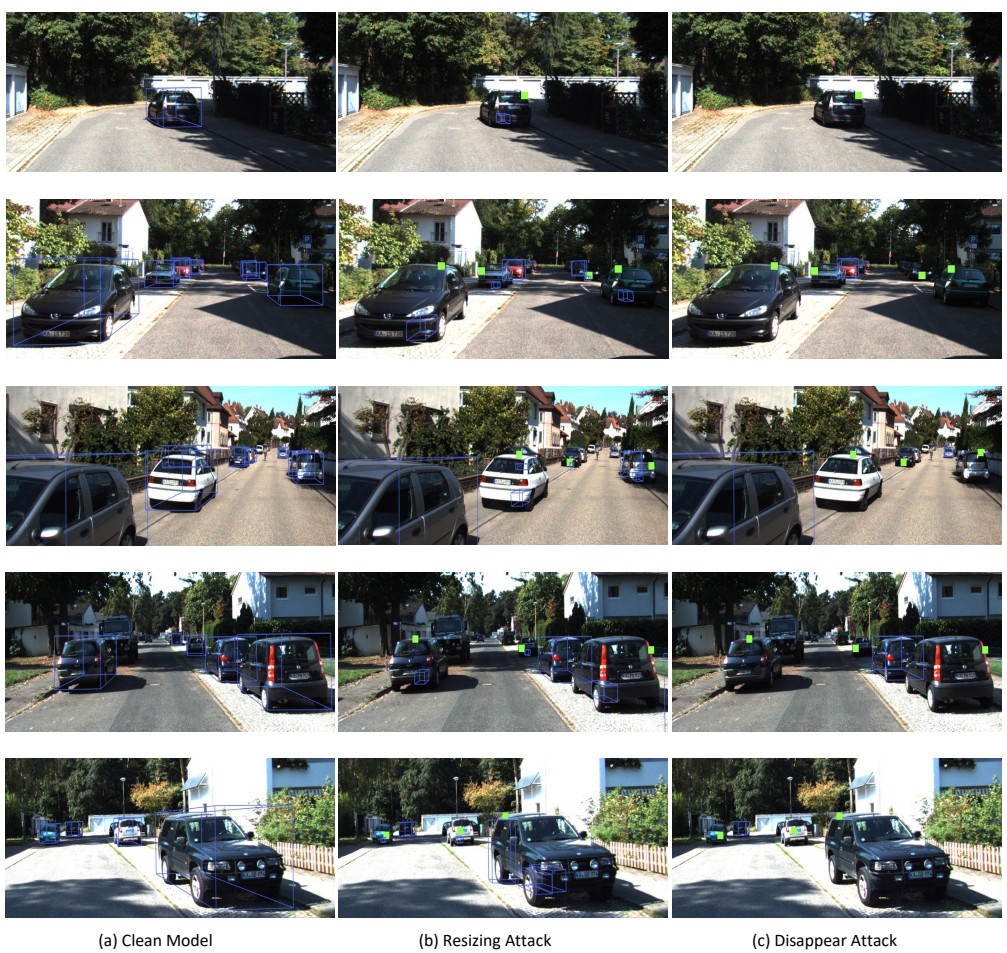

(a) Clean Model        (b) Resizing Attack        (c) Disappear Attack

Figure 8: Detection results of the clean model and BadFusion with different attack goals.

