# OpenReview forum: "BadFusion: 2D-Oriented Backdoor Attacks against 3D Object Detection"
_NeurIPS.cc/2023/Workshop/BUGS — NeurIPS 2023 BUGS Poster_

### Official Review · Reviewer_ac1Q · 2023-10-20
**An interesting paper addressing important issues**

**Rating:** 8
**Confidence:** 4

**Review:**

This paper explores how to design 2D backdoor attacks against 3D object detection. The authors demonstrate why we need to design 2D backdoor attacks against 3D object detection and discuss why existing 2D methods will fail in attacking 3D object detection. Based on these analyses, the authors propose the first effective method with fusion-aware and LiDAR-Free trigger design. The authors evaluate their method on KITTI with 4 baseline attacks.

In general, I enjoy reading this paper, especially its analyses of why we need to design 2D backdoor attacks against 3D object detection and why existing 2D methods will fail in attacking 3D object detection. The idea is well-motivated and effective. However, I still have some comments to help authors further improve their paper.

1. The current reference format is strange. Please double check before submitting the final version.
2. It would be better if the authors can evaluate their method also on other datasets.
3. It would be better if the authors can discuss the resistance of the proposed attack to potential (adapative) backdoor defenses.

---

### Official Review · Reviewer_VLiN · 2023-10-26
**A simple yet effective dock-door attack method proposed for a specific problem**

**Rating:** 6
**Confidence:** 4

**Review:**

This paper mainly discusses how to introduce a backdoor attack in the task of 3D object detection for autonomous driving, specifically in the fusion of 2D camera and 3D LiDAR. The author focuses on the fusion setting where 2D inputs are projected into 3D space to augment 3D signals. In this setting, the trigger is inserted into the 2D image, and when the 2D image is finally projected into 3D space, issues such as trigger sparsity and trigger inconsistency can easily occur, leading to the failure of the backdoor attack. In this paper, the author proposes using dense and solid-colored triggers for the backdoor attack.

Overall, this paper presents a method for backdoor attack in a specific scenario. Although the proposed method is relatively simple, the experimental results seem to show significant improvements. In summary, it can be considered a relatively good workshop paper.

---

### Decision · Program_Chairs · 2023-10-28

**Decision:**

Accept (Poster)

**Comment:**

Thank you for submitting to our workshop. Both the reviewers found the paper's contribution significant enough to be discussed at the workshop.